# Human rhinovirus promotes STING trafficking to replication organelles to promote viral replication

Martha Triantafilou [1,2✉], Joshi Ramanjulu[3], Lee M. Booty [1], Gisela Jimenez-Duran[1,2], Hakan Keles [4], Ken Saunders [5], Neysa Nevins[3], Emma Koppe[1], Louise K. Modis[5], G. Scott Pesiridis[3], John Bertin[3] & Kathy Triantafilou[1,2]

Human rhinovirus (HRV), like coronavirus (HCoV), are positive-strand RNA viruses that cause both upper and lower respiratory tract illness, with their replication facilitated by concentrating RNA-synthesizing machinery in intracellular compartments made of modified host membranes, referred to as replication organelles (ROs). Here we report a non-canonical, essential function for stimulator of interferon genes (STING) during HRV infections. While the canonical function of STING is to detect cytosolic DNA and activate inflammatory responses, HRV infection triggers the release of STIM1-bound STING in the ER by lowering $Ca^{2+}$, thereby allowing STING to interact with phosphatidylinositol 4-phosphate (PI4P) and traffic to ROs to facilitates viral replication and transmission via autophagy. Our results thus hint a critical function of STING in HRV viral replication and transmission, with possible implications for other RO-mediated RNA viruses.

[1] Immunology Catalyst, Immunology Network, GlaxoSmithKline, Gunnels Wood Road, Stevenage, Hertfordshire SG1 2NY, UK. [2] Cardiff University, Institute of Infection and Immunity, School of Medicine, University Hospital of Wales, Heath Park, Cardiff CF14, UK. [3] Innate Immunity Research Unit, GlaxoSmithKline, 1250 South Collegeville Road, Collegeville, PA 19426, USA. [4] Functional Genomics, GlaxoSmithKline, Gunnels Wood Road, Stevenage, Hertfordshire SG1 2NY, UK. [5] Adaptive Immunity Research Unit, GlaxoSmithKline, Gunnels Wood Road, Stevenage, Hertfordshire SG1 2NY, UK. ✉email: TriantafilouM@cardiff.ac.uk

Human rhinoviruses (RVs) are medically important single-stranded RNA viruses that belong to the family *Picornaviridae*. Originally neglected, as they were considered less virulent and capable only of causing mild common cold symptoms. It is only recently that they have been recognised to be important causative agents of acute upper and lower respiratory infections, wheezing, pneumonia, bronchiolitis, exacerbations of both asthma and COPD[1–5], as well as severe respiratory disease[6]. Together with coronaviruses, RVs are responsible for the majority of upper and lower respiratory tract infections in all age groups[7–9] as well as being a major cause of morbidity in infants, young children and the elderly[9,10]. In the United States alone, HRV causes 500 million non-influenza viral respiratory infections, resulting in a huge socioeconomic burden with estimated direct costs of USD $17 billion and indirect costs of $22.5 billion annually[11].

To date, there are 171 rhinovirus (RV) genotypes recognized and classified into the three species as RV-A (83 types), RV-B (32 types), and RV-C (56 types)[12–14]. The large number of RV serotypes and their considerable genetic diversity has represented a major obstacle to the development of antiviral agents[15], as well as a successful vaccine. Strategies to combat HRV infections and associated pathology are limited to supportive therapy and novel avenues are much needed. To date, there is no approved antiviral treatment for RV infections.

For HRV viruses to replicate and amplify their genomes, they reorganise intracellular membranes into membranous structures forming phosphatidylinositol 4-phosphate (PI4P) lipid enriched organelles termed replication organelles (ROs)[16,17].

Stimulator of interferon genes (STING) is an endoplasmic reticulum (ER)-associated membrane protein that has been shown to have many functions including sensing of intracellular DNA via the cGAS-STING pathway, resulting in triggering an interferon (IFN) and inflammatory response[18–21], as well as inducing autophagy for clearance of cytosolic DNA through a TANK binding kinase (TBK1) independent mechanism[22]. Although part of the DNA sensing machinery STING has been previously shown to be implicated in responses against RNA viruses[23]. It has been shown that cells lacking STING are more sensitive to RNA virus infection and that during those infections, STING plays a role in inhibiting the translation machinery to limit the RNA viral synthesis[24]. Crosstalk between RNA RLRs, such as RIG-I and MDA5, and STING has also been implied from studies with HIV-1[25,26], HTLV-1[27], and Dengue[28,29]. To date, no direct interaction of STING with foreign RNA has been reported.

STING activation seems to rely on its location within the cell. When STING localizes to the ER and is bound by stromal interaction molecule 1 (STIM1) it is rendered inactive[30]. Upon activation, STING traffics from the ER to the ER–Golgi intermediate compartment (ERGIC), where STING-mediated TBK1/ Interferon Regulatory Factor 3 (IRF3) signalling occurs[31]. STING activation needs to be regulated and excessive STING signalling is implicated in severe sterile inflammatory conditions[31,32]. Thus, STIM1 regulates STING trafficking and activation by directly interacting with STING to retain it in an inactive state on the ER membrane. STIM1 is an ER calcium sensor and when the ER $Ca^{2+}$ stores are depleted, STIM1 moves to the ER–plasma membrane (PM) junction sites, where it binds to calcium release-activated calcium channel protein 1 (CRCM1 (*Orai1*)) and opens $Ca^{2+}$ release-activated $Ca^{2+}$ (CRAC) channel to allow $Ca^{2+}$ influx[30]. The small hydrophobic HRV 2B protein targets the ER and the Golgi complex to reduce $Ca^{2+}$ levels[33,34]. Here, we have discovered that during HRV infection, HRV 2B reduces the ER and Golgi complex stored $Ca^{2+}$ forcing STIM1 to leave the ER, thus releasing STING. Once released, STING can bind PI4P via a patch of positively charged residues and is recruited to the RO, inhibiting downstream immune responses. STING can bind to phosphatidylserine (PS) autophagosomal vesicles which provide a cellular exit point for the virus, elucidating a role for STING as a homeostatic regulator between PI4P/PS lipid exchange. This study highlights a critical role for STING in HRV viral replication and transmission, involved in lipid recognition and homeostasis, with possible implications for other RO-mediated viruses, such as coronavirus.

## Results

**STING is required for HRV replication.** The identification of host proteins and pathways exploited by viruses to enhance their infectious cycle has historically provided insight into host–pathogen interactions and possible therapeutic avenues. Using an unbiased genome-wide small interference RNA (siRNA) screen of HRV infection in airway epithelial cells, *Tmem173* (STING) was identified as a strong cellular target of HRV, the only gene to have all four siRNAs within the top 30 hits, with two in the top ten[35].

To validate the requirement of STING in HRV replication, *Tmem173* (hereby referred to as STING) knockouts were generated by CRISPR-Cas9 editing in both BEAS-2B and air-liquid interface primary human airway epithelial (ALI) cells. STING-deficient cells were infected with A and B groups (including major and minor serotypes) such as HRV-A1B, HRV-A2, HRV-A16, HRV-B14, and HRV-B4 as well as other RNA respiratory viruses such as Respiratory syncytial virus (RSV) or Influenza A virus (IFV-A). This confirmed that STING downregulation abrogated HRV infection in all serotypes whereas RSV and IFV-A infection was not affected (Fig. 1a).

HRV group C (HRV-C), a novel, more pathogenic rhinovirus group which has been implicated in childhood acute asthma and causes more severe attacks than other viruses or HRV groups[36], was also investigated. Using the same ALI cultures as above, we infected wildtype or STING-deficient cells with HRV-C15 and assessed viral replication by qPCR over time. Our results confirmed that rhinoviruses HRV-C15 replication was also STING dependent (Fig. 1b).

Since our data suggested that STING is crucial for HRV replication in lung-derived cells, we proceeded to investigate STING expression in lung tissue. Profiling of mRNA expression in the GTEx database demonstrated detectable levels of STING across multiple tissue types, with greatest levels observed in lung tissue and blood (Fig. 1c). Expression in lung tissue is consistent with GSK internal microarray data from multiple studies (unpublished), which demonstrate consistent expression in isolated air-liquid-interface (ALI) cultured bronchial epithelial cells. In addition, transcriptomics analysis of STING expression in ALI cultures, demonstrated that STING was significantly upregulated in response to HRV infection (Fig. 1d).

**Canonical function of STING is compromised upon HRV infection.** It is well documented that in the presence of ligands such as interferon stimulatory DNAs (ISDs) as well as cGAMP, STING becomes activated, followed by TANK-binding kinase 1 (TBK1) activation and IRF3 phosphorylation leading to IFN-β production. Although HRV is an RNA virus, we investigated whether there was interferon-β (IFN-β) production in response to HRV infection in ALI cells. It was shown that ALI cells produced IFN-β in response to ISDs and cGAMP, but also HRV infection (Fig. 2a). However, BX795, a TBK1 inhibitor, as well as RU.521, a cGAS inhibitor, did not inhibit the HRV-induced IFN-β production (Fig. 2a), demonstrating cGAS-STING-independent IFN-β production post HRV infection.

As STING canonically cannot detect RNA, we assessed the possibility that HRV could be triggering cGAMP production, thus the level of cGAMP after infection with HRV, or with stimulation

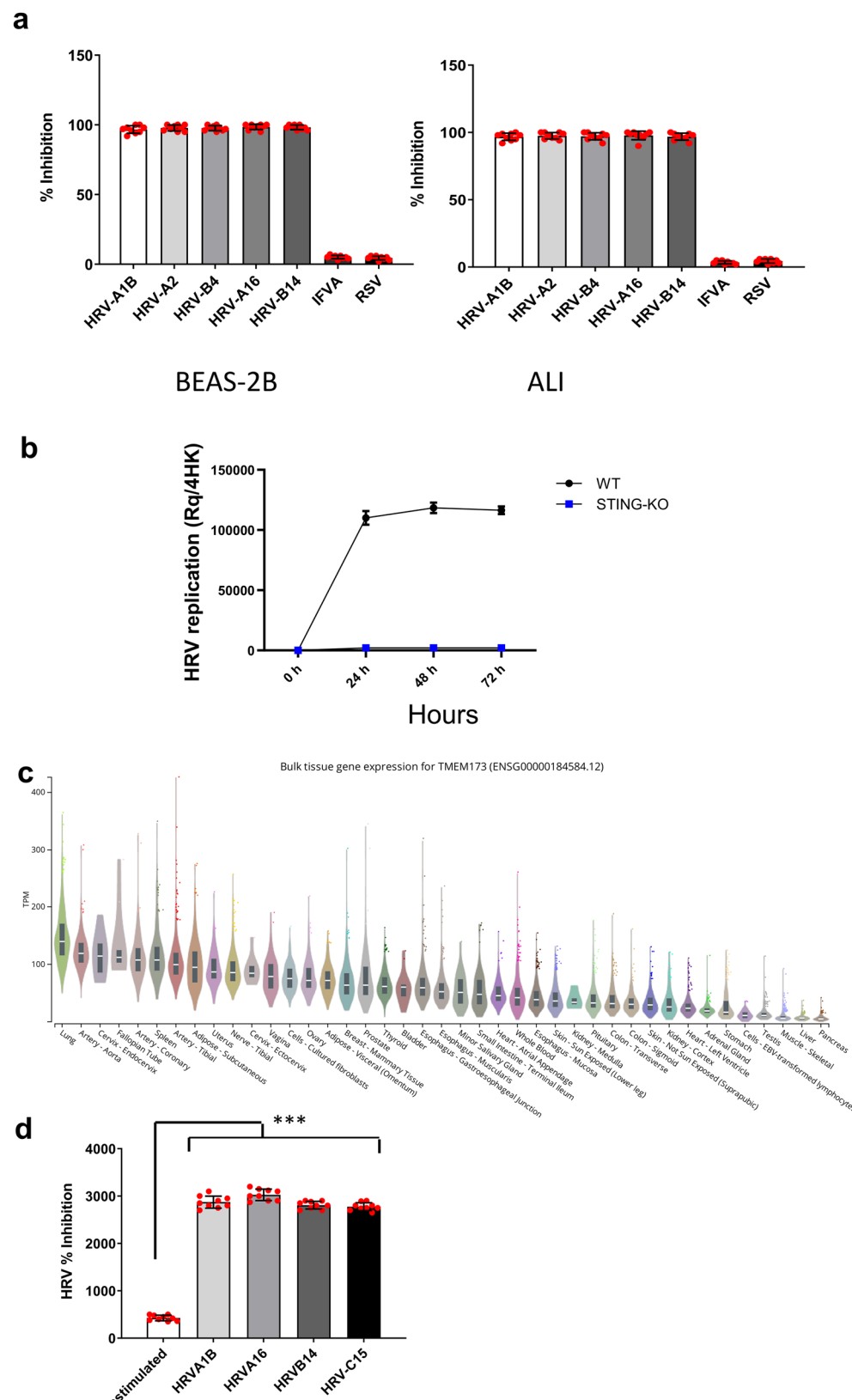

using ISDs was examined (Fig. 2b). Our data showed there was no production of cGAMP downstream of HRV infection, therefore aligning with inhibitor-obtained data above suggesting cGAS-STING independent IFN-β production.

Our group and others have shown IFN production upon HRV infection is RNA helicase dependent[37]. To show this, as well as STING-independence, we knocked out MDA5 and RIG-I using CRISPR. Whilst STING-deficient and wildtype cells were able to generate an IFN-response, MDA5/RIG-I KO cells were not (Fig. 2c). Additionally, HRV infection did not induce TBK1 phosphorylation, highlighting the independence from the STING-TBK1 pathway (Fig. 2d). Additionally, ablation of TBK1

**Fig. 1 STING is required for HRV replication.** BEAS-2B/STING knockout (KO) clones and airway epithelial (ALI)/STING KO clones were infected with HRV-A1B, HRV-B2, HRV-A4, HRV-A16, and HRV-B14 as well as other RNA respiratory viruses such as Respiratory syncytial virus (RSV) or Influenza A virus (IFV-A) as controls. Viral replication was assessed using qPCR and percentage inhibition of virus replication is depicted. Data are means ± SD ($n = 3$, from three independent experiments) (**a**). ALI STING KO were also infected with HRV-C15. Data are means ± SD ($n = 3$, from three independent experiments) (**b**). STING mRNA tissue expression data from the GTEx database plotted as transcripts per million (TPM) on a linear scale, excluding results from brain (all with median TPM below 10) and ranked by median expression value per tissue. Boxplots show median and 25th and 75th percentiles. Points are displayed as outliers if beyond 1.5 times the interquartile range (**c**). STING RNA expression before and after infection with HRV-A1B, HRV-A16, HRV-B14, and HRV-C15. Data representative of three independent experiments. Statistical significance between unstimulated and virus-infected conditions was assessed by unpaired student's $t$-test. Data is mean ± SD ($n = 3$). ***$p < 0.001$ indicate statistically significant (**d**).

or IRF3 had no effect on HRV replication in bronchial epithelial cells as measured by qPCR (Fig. 2e). As expected, these results collectively indicate that RLRs, and not STING, drive the IFN-response to HRV infection. Interestingly, when ALI cells were challenged with HRV followed by cGAMP or ISDs, STING was unable to signal as expected with no TBK1 or IRF3 phosphorylation (Fig. 2f). These results collectively indicate that RLRs, and not STING, are responsible for downstream IFN production following HRV infection in respiratory cells, and more interestingly that STING cannot canonically function following HRV infection.

**STING traffics to PI4P-rich viral replication organelles during early infection.** One important feature of STING signalling is its trafficking. STING localizes to the ER in the resting state and upon activation, STING moves to the ER–Golgi intermediate compartment (ERGIC), and then to the Golgi where it triggers phosphorylation of TBK1 and IRF3 at the trans Golgi network (TGN)[38]. STING is also post-translationally palmitoylated at the TGN, as is required for IFN response[39].

Studies have confirmed that blocking the ER-Golgi route for STING using brefeldin A inhibits STING-dependent signalling, confirming that ER-Golgi traffic is detrimental for STING activation[11]. Coupled with our observations of interference by HRV upon traditional STING signalling, we wanted to examine the intracellular behavior of STING upon HRV infection. Cells were infected with HRV, and STING trafficking was monitored by confocal microscopy. Our results showed that STING did not follow the canonical ER to ERGIC route upon infection, unlike stimulation with cGAMP (Fig. 3a).

Instead, in HRV-infected cells STING was targeted to vesicular structures. Similar to other positive-strand RNA viruses, HRV replicates in ROs forming distinct vesicular clusters from cytoplasmic membranes, composed of host proteins and lipids[40]. We infected bronchial epithelial cells with HRV-A1B and monitored STING trafficking post-infection (p.i.) throughout the virus replicative cycle (6–8 h). Our data showed that at 2–4 h p.i. STING trafficked to the viral ROs, and colocalised with the virus replicating dsRNA as well as the HRV replication protein 3A (Fig. 3b) and PI4P (Supplemental Fig. 1).

To determine the lipid content and composition of the replication organelles we used Coherent Raman scattering microscopy (CRS) techniques. Stimulated Raman scattering (SRS) and Coherent anti-Stokes Raman Scattering (CARS), are complementary to fluorescence modalities. They offer non-invasively intrinsic chemical selectivity without staining, as different molecules have specific and detectable vibrational motion[41].

SRS microscopy is a relatively new approach combining two short-pulsed near-infrared laser sources and by overlaying them both spatially and temporarily at the sample it is possible to coherently probe the vibrational signature of molecules with a spatial resolution on sub-micron length scales. Thus, we used SRS to identify the distribution and identification of lipids in the ROs.

Our results showed that STING-positive ROs had an increase in lipid composition, particularly phosphatidylinositol-4-phosphate (PI4P) (Fig. 3c). This is in agreement with previous studies which have shown ROs in picornavirus and hepacivirus are enriched in PI4P[42].

To test whether other HRV serotypes had the same replication pattern, ALI cultures were infected with HRVC15 and imaged using fluorescence microscopy. The data confirmed as expected that dsRNA from the virus was colocalising with PI4P during the replication cycle (Fig. 3d).

Phosphoinositol-4-kinase IIIα (PI4KIIIα) and phosphoinositol-4-kinase IIIβ (PI4KIIIβ) are responsible for the production of PI4P from phosphatidylinositol (PI) and are important for the replication of other RO-mediated RNA viruses, such as HCoV and poliovirus[42–44]. To determine whether PI4K was required for HRV infection, we firstly investigated whether PI4KIIIα or PI4KIIIβ co-localize with ROs and the HRV dsRNA. As expected, our data showed colocalization of PI4KIIIβ with the dsRNA associated with the RO upon HRV infection (Supplemental Fig. 1e). To assess this further, we utilised the GSK inhibitors GSK2998533, a PI4KIIIβ antagonist, and GSK268449 and PIK93, PI4KIIIα antagonists, in both confocal microscopy (Supplemental Fig. 1e) and viral replication assays (Supplemental Fig. 1f). GSK3000140 was used as an inactive negative control compound. It was clear that inhibition of PI4KIIIβ strongly reduced colocalization of dsRNA to STING, as well as diminishing viral replication.

In order to determine whether PI4KIIIβ interacts with STING and is required to recruit STING to the replication organelles, we performed fluorescence resonance energy transfer (FRET) studies to determine STING interactions with PI4KIIIβ, PI4P and ER (Fig. 4a, b). STING was shown to interact with PI4KIIIβ and PI4P in the ER 1 h after HRV infection, suggesting that PI4KIIIβ initially interacts with STING in the ER before translocating to the ROs.

These results demonstrate that HRV infection does not drive STING to travel to the ERGIC nor the TGN, where the palmitoylation of STING typically occurs. Therefore, this aberrant trafficking could explain why its canonical function and subsequent downstream signalling is compromised during HRV infection as observed in Fig. 3. To confirm this hypothesis, we investigated whether palmitoylation of STING still occurred after its relocation to the RO following HRV infection.

To test whether STING is palmitoylated, we utilized the in vitro Acyl-Biotin exchange (ABE) assay to determine the palmitoyl level of STING in HRV infected cells. As expected, it was shown that STING becomes palmitoylated only when cells are stimulated with ISDs, while in HRV infected cells STING is not palmitoylated (Fig. 4d). Thus, confirming that location is important for STING palmitoylation and the localization of STING at the ROs during HRV infection prohibits its palmitoylation and limits its subsequent signalling.

**HRV 2B protein reduces ER Ca$^{2+}$ levels and triggers release of STING from the ER.** Our data confirmed that HRV selectively

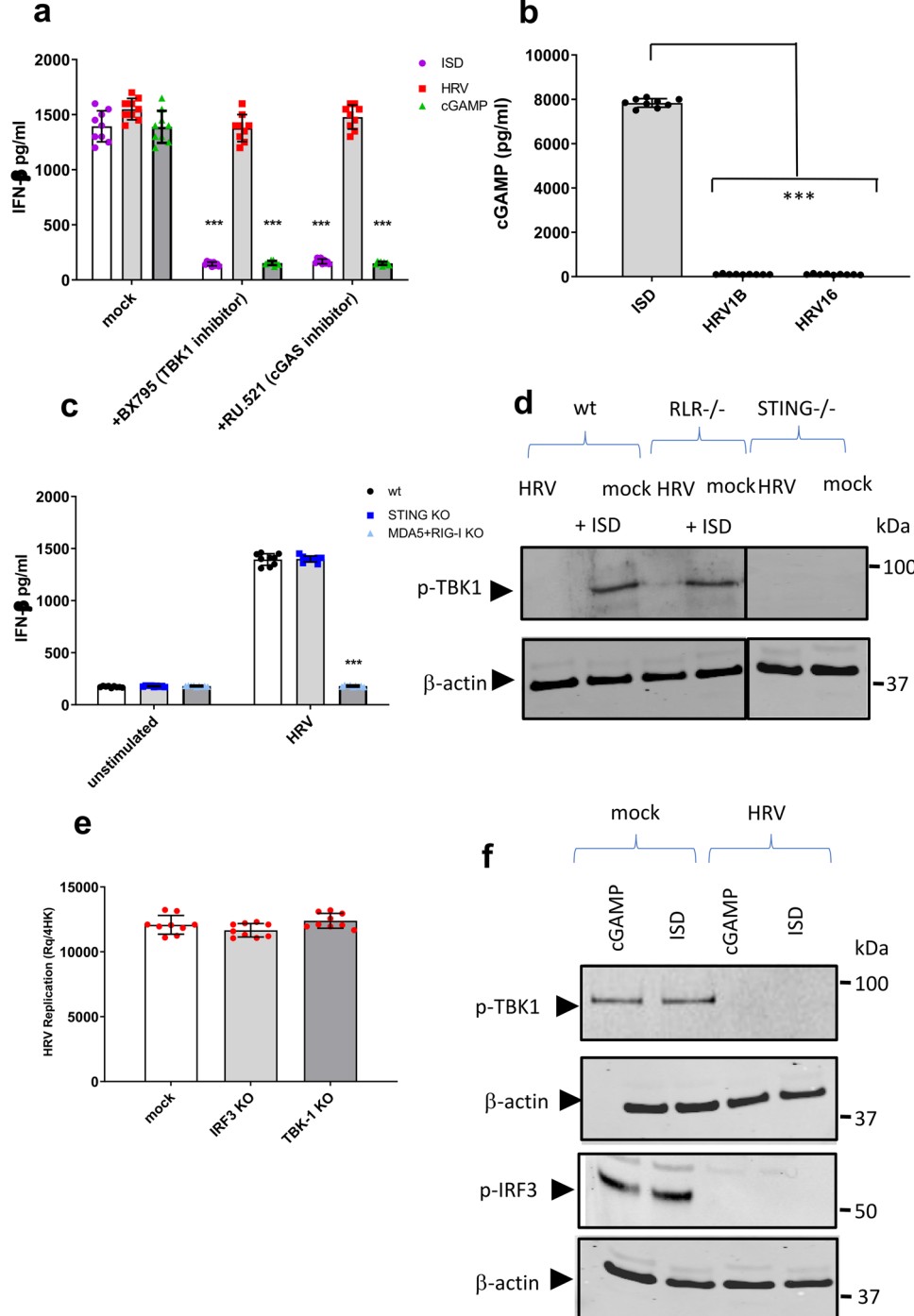

**Fig. 2 STING role in HRV infection is independent of cGAS/STING canonical pathway.** IFN-β production in BEAS-2B cells infected with HRV-A1B in the presence of inhibitors BX795 (1 µM) and RU.521 (500 nM) for 24 h measured by ELISA. Data are means ± SD (*n* = 3, from three independent experiments). ***p* < 0.001 indicate statistically significant (**a**). cGAMP levels in bronchial epithelial cells infected with different strains of HRV (MOI 1) was measured by ELISA. Data are means ± SD (*n* = 3, from three independent experiments). ***p* < 0.001 indicate statistically significant (**b**). BEAS-2B (WT), BEAS-2B STING-deficient cells (STING KO) and BEAS-2B deficient cells for MDA5 and RIG-I (MDA5 + RIG-I KO) were infected with HRV-A1B. IFN- β levels in the supernatant after 24 h were measured by ELISA. The data represent the mean of three independent experiments ± SD (*n* = 3) yielding consistent results. Statistical significance between unstimulated and virus-infected conditions was assessed by unpaired student's *t*-test **p* < 0.005 and ***p* < 0.001 (**c**). The presence of p-TBK1 in lysates from BEAS-2B wildtype, STING KO or RLR KO BEAS-2Bs infected with HRV-A1B (MOI 1) for 1 h or exposed to 1 µg ISDs was detected via Western blotting (*n* = 3 independent experiments, representative western blot shown) (**d**). IRF3 or TBK1 was also knocked out in bronchial epithelial cells to determine whether it had an effect on HRV inhibition. Data are means ± SD (*n* = 3, from three independent experiments) (**e**). Bronchial epithelial cells infected with HRV-A1B (MOI 1) as well as mock (uninfected) cells were stimulated with either 1 µg ISD or 5 µg cGAMP, after 24 h cells were lysed and the presence of p-TBK1 and p-IRF3 were detected via western blotting (*n* = 3 independent experiments, representative western blot shown). Data are means ± SD (*n* = 3), ***p* < 0.001 (**f**).

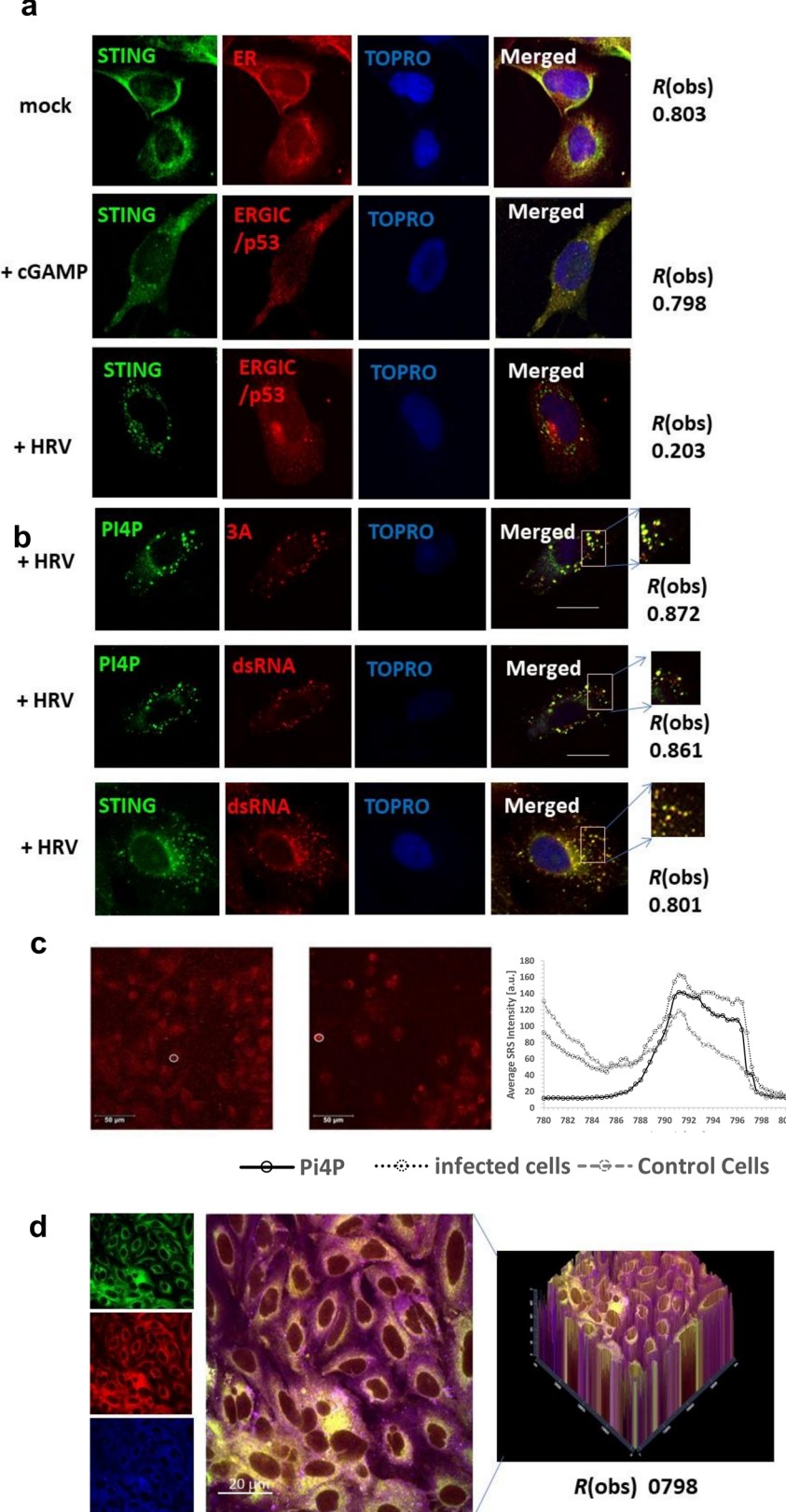

**Fig. 3 STING traffics to PI4P rich replication organelles.** Fluorescent micrographs of STING localization to ERGIC or ER in BEAS-2B cells following exposure to cGAMP (5 μg) or viral infection by HRV-A1B (**a**). Fluorescent micrographs of PI4P and STING localization with HRV 3A or dsRNA after HRV infection (**b**). Bars shown are 10 μm and data presented are representative images of $n = 3$ biological replicates, with at least 20 technical repeats. Degree of colocalization is shown as R(obs) for that image. Stimulated Raman Scattering (SRS) images showing total lipid distribution (796 nm) of control (left) and infected cells (center). Averaged SRS spectra extracted from regions of interest (ROs) indicated in the images with white circles overlaid on to PI4P lipid spectrum (**c**). ALI cultures infected with HRVC15 were subject to colocalization analysis between STING, PI4P and dsRNA (**d**).

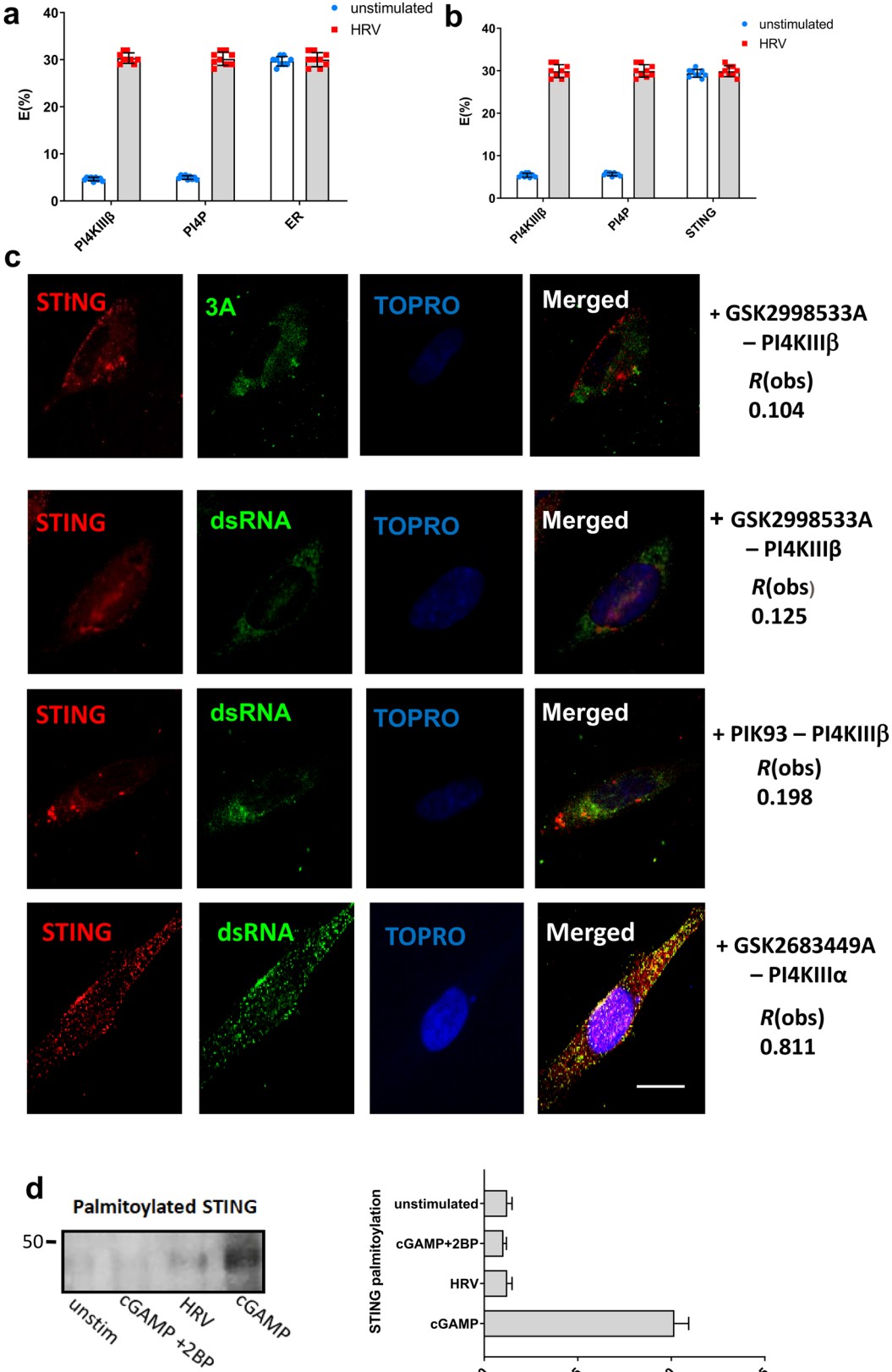

**Fig. 4 STING interactions with PI4KIIIβ, PI4P and ER.** Fluorescence resonance energy transfer (FRET) studies to determine STING interactions with PI4KIIIβ, PI4P and ER (**a**), as well as ER interactions with PI4KIIIβ, PI4P and STING (**b**) in BEAS-2B cells infected with HRV for 1 h. Data is means $+/-$ SD ($n = 3$, independent experiments). BEAS-2B cells infected with HRV-A1B (MOI 10) in the presence of GSK2998533 (100 nM), GSK2683449 (100 nM) or PIK93 (1 μM) and co-localization of STING, 3A and dsRNA was assessed (**c**). BEAS-2B cells untreated or treated with cGAMP (5 μg/ml) or infected with HRV (for 4 h) were subjected to Acyl-Biotin Exchange assay (ABE) assay, western blot analysis for STING palmitoylation. Cells treated with palmitoylation inhibitor 2 bromo palmitate (2BP) at 8 μm for 6 h is also shown (**d**). All data in A-H is shown as a representative image of three biological replicates that included at least 20 technical replicates. Data in G is mean $+/-$ SD ($n = 3$, independent experiments).

forms specialized ROs composed of cellular phosphoinositide lipids and that STING traffics to those ROs during HRV replication. However, the mechanism initiating this trafficking is unclear. In its resting state STING localizes to the ER membrane and it was recently shown by Srikanth et al. that stromal interaction molecule 1 (STIM1), a $Ca^{2+}$ sensor, interacts with STING to retain it to the ER membrane. STIM1, is a $Ca^{2+}$ binding protein which localizes throughout the ER when [$Ca^{2+}$] ER is high, but, after depletion of the ER $Ca^{2+}$ stores, it translocates into junctional areas between the ER and plasma membrane and interacts with the pore subunit of store-operated $Ca^{2+}$ channels, CRCM1, and induces $Ca^{2+}$ entry[30].

During HRV infection STING is not ER resident nor bound by STIM1 but instead moves to the replication organelles, implying [$Ca^{2+}$] ER has been depleted allowing the release of STING. The HRV 2B protein is a small hydrophobic protein that has been shown to localize to the ER-Golgi complex and reduce [$Ca^{2+}$] ER levels[34]. We hypothesized that during viral infection HRV 2B reduces the ER $Ca^{2+}$ stores causing STIM1 to translocate into junctional areas and thus releasing STING.

Investigation of both cytosolic and ER $Ca^{2+}$ levels during HRV infection demonstrated that HRV 2B protein reduces the $Ca^{2+}$ content of the ER (Supplemental Fig. 2). Confocal imaging data also showed that upon HRV 2B protein expression and subsequent ER $Ca^{2+}$ depletion, STIM1 trafficked away from the ER (Supplemental Fig. S2).

Fluorescence resonance energy transfer (FRET) was used to determine STING and STIM1 interactions. FRET can occur over 1–10 nm distances, and effectively increases the resolution of light microscopy to the molecular level. The data confirmed that STIM1 and STING interacted in uninfected cells (Fig. 5a). However, upon HRV infection there was no interaction between STIM1 and STING. This data, therefore confirms that HRV 2B affects the cellular $Ca^{2+}$ homeostasis and triggers STING release from STIM (Fig. 5a, b).

**STING binds to PI4P via positively charged amino acids**. To elucidate the key drivers for STING recruitment to the RO, we investigated the relationship between STING and different phospholipids during HRV infection. It has been shown that changes in cellular lipid and phospholipid content occur during PV infection[40], which belongs to the same family as HRV.

Using a lipid array, we found that STING could bind to phosphatidylinositol 4-phosphate (PI4P), phosphatidylinositol-4,5-bisphosphate (PIP2), and phosphatidylinositol (PI) thus highlighting possible STING-phospholipid interactions (Fig. 6a). The specificity of STING for phosphatidylinositol phosphates was further verified by measuring FRET interactions with these lipids (Fig. 6b).

Protein–membrane interaction is regulated by electrostatic interactions between the negatively charged phospholipids of the plasma membrane and the positively charged amino acids in a domain of the membrane-binding protein. Therefore, we hypothesized that STING interacted with these lipids in a similar manner. Using the MOE Site Finder[45], we identified a region of conserved positively charged amino acids on the α3 helix of STING between 280aa and 295aa that contains four evolutionary conserved positively-charged amino acids (Fig. 6c).

The STING N-terminus affords several sites with high protein–ligand binding (PLB) propensity scores[46] that could accommodate larger molecules such as a steroid or phosphatidylinositol phosphate. Site finder identified alpha spheres that can occupy protein surface concavities and scored them by proximity to amino acid residues. White and red alpha spheres are near hydrophobic and hydrophilic residues, respectively (Fig. 6c). The

residues R294 and R298 (K289 and R293 in human) are proximal to STING's N-terminal tail and thus less likely to interact with the PI4P phosphate, which could more easily interact with R286 or R289 (corresponding to human R281 or R284, respectively), especially when STING is in an active state (Fig. 6d).

To test whether these residues are involved in PI4P recruitment, we mutated all four to alanine (4posA) or glutamate (4posE). When ectopically expressed in STING deficient huh7.0 cells, we observed using confocal microscopy that both the STING 4posA and 4posE mutants could no longer interact with PI4P (Supplemental Fig. 3a), this was also confirmed using the lipid array strips (Fig. 6a) where STING 4posA and 4posE were shown not to be able to bind to PI4P on the strip. These results confirm that STING's "positive patch" serves as a PI4P binding domain for STING recruitment to the ROs.

Additionally, the use of H-151[47], a STING antagonist, was able to completely inhibit HRV infection as well as limit the interaction between STING and PI4P (Supplemental Fig. 3c).

This data suggests that upon HRV infection, STING is released from STIM1 due to alterations in calcium homeostasis, which drives the binding of STING to PI4KIIIβ and PI4P and recruitment to the RO. Using both genetic modifications and chemical interventions, it was possible to limit this interaction, stalling STING in the ER and limiting HRV replication (Supplemental Fig. 3d).

**Upon completion of replication STING traffics in phosphatidylserine enriched autophagosomes**. Recent studies on sea anemone *Nematostella vectensis* and *Xenopus laevis* have shown that STING is important for autophagy induction[22]. Upon binding cGAMP, STING translocates to ERGIC which serves as a membrane source of LC3 lipidation and formation of autophagosomes[22]. However, we have seen that in HRV infected cells STING does not travel to ERGIC but binds PI4P and traffics to ROs instead.

Multiple studies have shown the accumulation of autophagosomes following picornavirus infection promoting non lytic release of picornavirus autophagosome derived vesicles[21]. Many pathogens are degraded by autophagy, but some subvert autophagy for their own advantage. Other *Picornaviridae* related to HRV, such as Coxsackievirus B3 (CBV3), induces the autophagosomal pathway but inhibits degradation of autophagic cargo[48].

HRV infectious cycle is typically from 6 to 8 hours, thus we imaged STING translocation at different times post infection (p.i.) by confocal microscopy. STING trafficked to PI4P rich ROs which became visible at 2–4 h p.i., then between 5 and 8 h p.i. STING sequentially moved from the ROs to autophagosome like vesicles forming large vesicular clusters that appeared after the peak of RNA replication (Supplemental Fig. 4).

We also investigated the timeframe of viral RNA replication as well as viral assembly and release. We infected bronchial epithelial cells with HRV and then at various times post-infection (p.i.) we measured viral RNA as well as infectious virus by plaque assay (Supplemental Fig. 3e, f). Viral replication peaked at 4–6 h p.i., while after the peak between 6 and 8 h p.i. there was an increase in extracellular viral titre, which corresponded with the autophagosome like vesicle formation.

The generation of HRV particles is a multistep process. Once RNA is packaged, the VP0 subunits get cleaved into VP2 and VP4 to generate mature infectious virions. To visualize the assembled capsids, we used an anti-VP2 antibody. Confocal imaging showed that between 5 and 8 h p.i. 80% of assembled virus capsids were in these vesicles as well as the autophagosomal marker LC3 and STING (Fig. 7a).

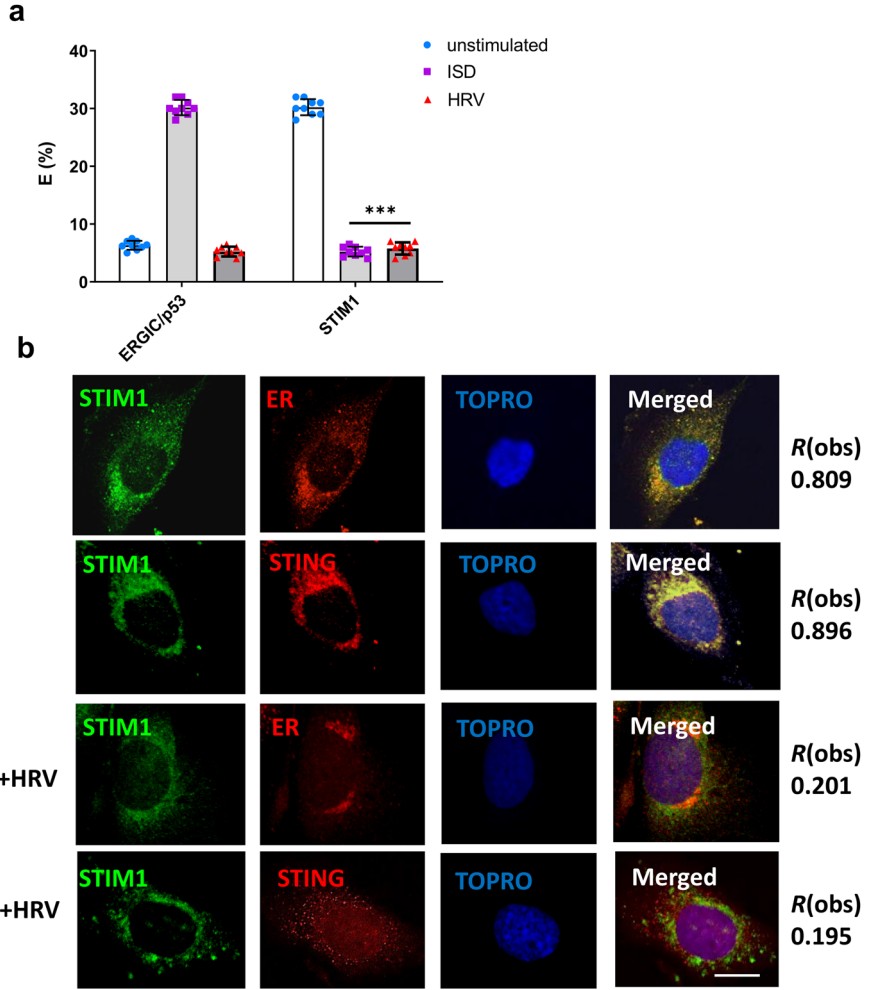

**Fig. 5 STIM1 releases STING upon HRV infection.** FRET studies measuring donor (STING) and acceptor (ERGIC or STIM1) interactions in BEAS-2B cells infected with HRV for 2 h or exposed to ISD (1 μg) (**a**). Confocal microscopy of BEAS-2B cells infected with HRV for 2 h assessing colocalization between STIM1, the ER and STING. Data in **a** is means +/− SD (n = 3, independent experiments). Statistical significance between unstimulated and virus-infected conditions was assessed by unpaired student's t-test. ***p < 0.001 Data in **b** is representative of three biological donors with at least 20 technical replicates. Degree of colocalization is presented at R(obs). Bars shown at 10 μm.

We tested for the presence of different phospholipids and the vesicles that appear after genome replication contained mainly negatively charged phosphatidylserine (PS) lipids and were distinct from ROs (Fig. 7b). Therefore, the earliest stages of PI4P induction and membrane remodelling actually represent the RO but these vesicles encase autophagy markers. High throughput fluorescence (HTF) imaging showed PS redistribution leading to more localised and intense spots/nodes of lipids in the PS rich organelles of the infected cells compared to that of the control cells where more uniform lipid distribution was observed (Supplemental Fig. 5). Furthermore, heat maps representing number of detected nuclei as shown also reveals that number of nuclei is noticeably increased at 6 h of post infection. This is expected since nuclei are more prominent in autophagy and HRV forms autophagosome like vesicles (Supplemental Fig. 5). Additionally, using Stochastic Optical Reconstruction Microscopy (STORM) we confirmed the presence of STING and LC3 in these PS rich vesicles that were larger than ROs ranging from 350 to 450 nm (Fig. 7b, c).

In order to determine whether STING plays a role in the virus package and release, ALI cells were given H-151, a STING antagonist, either as a pre-treatment or at the middle of the HRV infectious lifecycle and extracellular HRV was determined at 8 h of post infection. It was shown that when H-151 was administered as a

pre-treatment, HRV replication was completely inhibited (Supplemental Fig. 3e, orange line graph) suggesting that STING plays a role in the replication of the virus. However, when H-151 was administered in the middle of the HRV replicative lifecycle and extracellular HRV was assessed 8 h of post infection, it was shown that H-151 could inhibit HRV release to the extracellular space suggesting that STING also plays a role in the virus package and release (Supplemental Fig. 3e, gray line graph).

From our experiments, we propose that two different organelles are induced during HRV infection that serve temporally distinct functions during HRV infectious lifecycle, and STING contributes to formation and function of both.

The membranes of eukaryotic cells vary in lipid composition, a feature essential to organelle identity. It has been shown that oxysterol binding related proteins ORP8 and ORP5 extract and exchange PI4P for PS between two membranes[49] and it is possible that STING, since it binds PI4P, could mediate the PI4P/PS lipid exchange between organelle membranes thus maintaining membrane lipid homeostasis. It is likely that the autophagosomal like organelle originated from ER derived replication organelle membranes.

Using differential ultracentrifugation followed by density gradient ultracentrifugation, we collected the cell fraction that

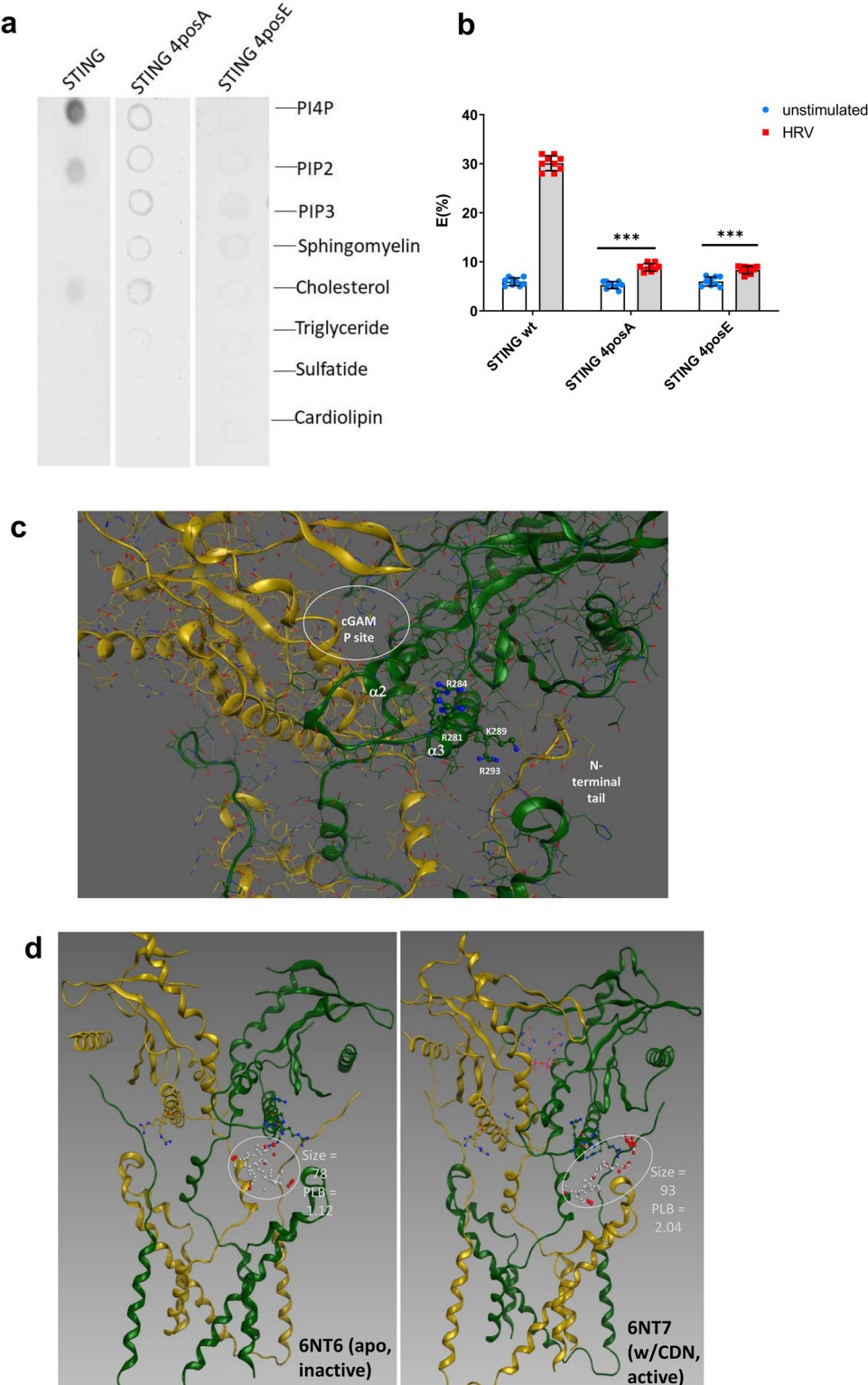

**Fig. 6 STING interacts with PI4P via a patch of positive charged amino acids.** Recombinant STING or purified STING 4posA or STING 4posE were incubated with a PIP strip expressing different phospholipids and binding was revealed by anti-STING antibody followed by the appropriate secondary antibody and visualisation (**a**). ($n = 3$ independent experiments, representative western blot shown). Huh7.0 wildtype, STING 4posA, or STING 4posE cells were infected with HRV for 4 h before FRET efficiency (E%) assessment for STING and PI4P. Data are means $+/-$ SD ($n = 3$, independent experiments). ***$p < 0.001$ indicate statistically significant (**b**). MOE Sitefinder alpha spheres indicate space for large ligand binding partners; residues comprising charged patch circled; alpha spheres: white (near hydrophobic residues) and red (near hydrophilic residues) (**c**); comparison of the charged patches in inactive (right) or active (left) STING (**d**).

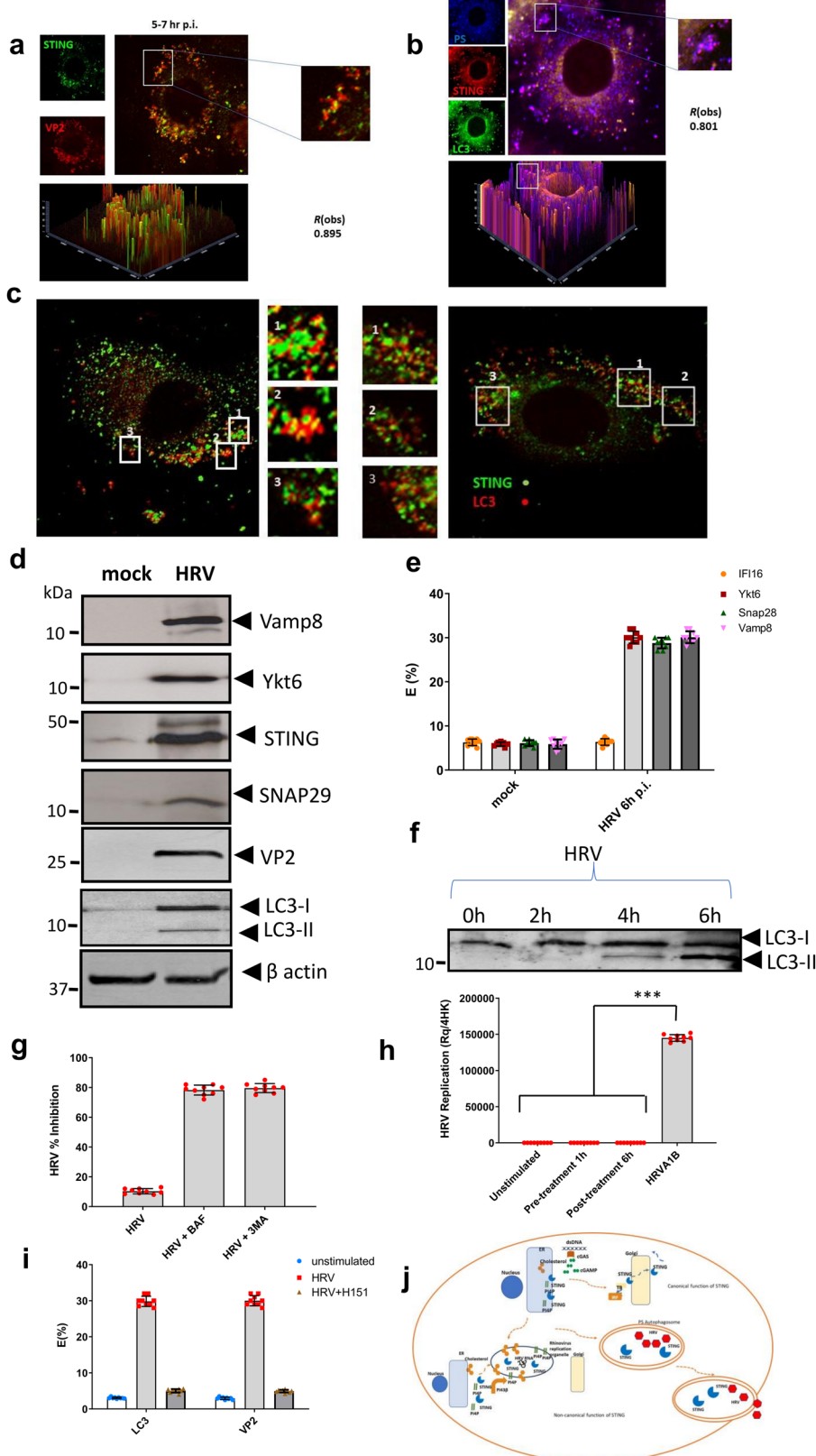

contained the autophagosome/PS rich vesicles. Magnetic beads conjugated to annexin V was used to further purify the vesicle fraction), since annexin V can bind phosphatidylserine, with clear presence of STING and LC3 with VP2 (Fig. 7).

Screening by western blotting for different SNARE proteins we identified that the Ykt6 protein was also present in these vesicles

(Fig. 6d). Ykt6 is a unique R-SNARE protein highly conserved from yeast to man localized to both the cytosol and intracellular membranes. It is an important regulator of the vesicular transport between the ER and the Golgi[50] and recently has been shown to be implicated in autophagosome biogenesis and fusion[51]. Thus, it seems that Ykt6 provides a reserve pool for the fusion and

**Fig. 7 STING travels to phosphatidylserine (PS) autophagosomes.** Bronchial epithelial cells infected with HRV were fixed 7 h p.i. and immunolabeled with VP2 (red) and STING (green) (region of interest depicting the formed vesicles is magnified on the right.) (**a**). Bronchial epithelial cells infected with HRV 7 h p.i. were immunolabeled with STING (red), PS (blue) and LC3 (green) (region of interest depicting the formed vesicles is magnified on the right) (**b**). The merged images show extensive colocalization. The data presented are at least in 15 cells over three independent experiments. The degree of colocalization R(obs) was determined using ImageJ software as Pearson's correlation coefficient (r) via the Costes' method. Bars shown at 10 μm. STORM superresolution imaging of STING and LC3 distribution in autophagosomal vesicles is also shown. Relative fluorescence of the target structures where regions of interest (ROI) were selected for the target areas are shown. The distribution/position of receptors was obtained with Zeiss Zen black or ImageJ software (**c**). Western blotting was performed to examine protein expression of STING, Ykt6, VP2 (virus assembly marker), LC3, and Vamp8 in the isolated cell fraction that contained the autophagosome/PS rich vesicles (n = 3 independent experiments, representative western blot shown) (**d**). FRET efficiency (E%) to determine receptor interactions is also shown between STING, Ykt6, snap29, and Vamp8 in mock and HRV infected cells 6 h p.i. (n = 3 independent experiments) (**e**). Furthermore, LC3 lipidation was monitored over a time course of HRV infection by western blotting (n = 3 independent experiments, representative western blot shown) (**f**). Bronchial epithelial cells were either mock or HRV infected for 6 h in the presence or absence of bafilomycin (BAF, 200 nM) or 3MA (10 mM). Viral titres were analysed by plaque assay and % HRV inhibition is shown (**g**). In addition, ALI cultures were either pre-treated or post-treated with STING antagonist and subsequently infected with HRV. Replication was assessed using qPCR 24 h of post infection. ***p < 0.001 indicate statistically significant (**h**). FRET studies measuring either donor (STING) and acceptor (LC3 or VP2) interactions in BEAS-2B cells infected with HRV for 6 h (**i**). Model figure of HRV-STING interactions created by KT (**j**). Data are represented as mean ± SD from three independent experiments.

membrane transport that leads to the formation of compartments such as the autophagosome-like vesicles during HRV infection. The presence of SNAP29 and VAMP8 was also detected (Fig. 7d).

Subsequently we examined the distribution of STING and Ykt6 before and after HRV infection using FRET. Following 5 and 8 h p.i. of HRV infection, STING assembled in these autophagosome-like vesicles enriched with PS lipids together with Ykt6, SNAP29, and VAMP8 (Fig. 7e).

**STING induces conversion of lipidated LC3 in the autophagosomal like vesicles.** To determine the effect of autophagosome on HRV infections and to elucidate the role that STING plays in the formation of this autophagosome like vesicle we looked at LC3 lipidation over the time course of the infectious cycle. Our data demonstrated an elevated conversion of LC3-I to LC3-II (Fig. 7f) between 4–6 h p.i. of the virus infectious cycle suggesting augmented autophagosome formation.

Bronchial epithelial cells infected with HRV were treated with autophagy inhibitor bafilomycinA1 (bafA1), which blocks autophagolysosome formation, as well as 3-Methyladenine (3- MA), an established inhibitor of autophagy. Treatment with bafA1 or 3-MA inhibited viral yield suggesting that autophagy has a role in HRV propagation (Fig. 7g).

Previous studies have shown a role for autophagy in non-lytic release of *Picornaviridae*. To determine the destination of these non-canonical autophagosomal vesicles, we used the cell impermeable Trypan blue dye between 7–8 h p.i. and found that the plasma membrane remained intact. We proceeded to perform time lapse differential fluorescent microscopy in the presence of Alexa488-Annexin V and observed numerous Annexin V labeled vesicles emerged from the plasma membrane (Supplemental Fig. 6)

We have seen that inhibitors of autophagy reduce viral load, and we propose a model by which these non-canonical autophagosome like vesicles provide a mechanism for nonlytic release of rhinovirus. LC3 was detected and co localized with STING and viral capsid protein VP2 demonstrating that these vesicles contain HRV assembled virions (Fig. 7a).

The use of STING antagonist H-151, which abrogated interactions with PI4P and retains STING in the ER completely inhibited virus production (Fig. 6h) and subsequent autophagosomal vesicle formation. FRET studies in the presence of STING antagonist H-151 showed that there was no interaction between STING and LC3 or STING and virus VP2 thus confirming that STING is essential for autophagosome formation which is used for virus assembly and release during HRV infection (Fig. 7i).

## Discussion

Since RVs have long been regarded to be the etiological agent of the common cold, a relatively simple, self-limiting syndrome, their importance as possible causal factor of severe respiratory illness has often been neglected. In the past few decades, the importance of RVs in upper and lower respiratory infections, pneumonias, wheezing, and exacerbations of asthma and COPD has become apparent. Despite its socioeconomic impact the broad variety of RV genotypes has represented a major obstacle in the development of specific vaccines or antiviral agents for RVs. In the current study, STING has been found to be crucial for viral replication across all RV serotypes, demonstrating a very novel therapeutic avenue.

STING sensing and activation is tightly controlled by a series of conformational changes, thresholds of activation as well as its location within the cell. In our study, we have identified STING as a multifaceted receptor and an important unexpected feature in HRV replication of all RV strains, with promise as a therapeutic target. A recent study has also confirmed this, but only for HRV-A and HRV-C, but not HRV-B strains[35]. This difference could possibly by due to the cell types that the authors have used in their study and the virus tropism, as the authors mainly used Huh (hepatic) cells as well as HeLa (cervical epithelial) cells to confirm their findings, whereas our study utilises biologically relevant cell types such as bronchial epithelial and ALI cells that are more susceptible to all respiratory viruses.

In our study, we observed that upon HRV infection, HRV-2B protein affects the intracellular $Ca^{2+}$ homeostasis thus STING–STIM1 interaction is disrupted (Supplemental Fig. 7), STING exits the ER and is recruited to the viral ROs through an interaction between a patch of positively charged amino acids on STING and PI4P, which allows recruitment of STING to ROs.

Using the MOE Site Finder, we identified a patch of conserved positively charged amino acids on the α3 helix of STING between 280aa and 295aa suggestive of a region where STING interacts with PI4P. In particular, the positive-charged residues R281 or R284, which reside in the polymer interface of STING seem to be crucial for allowing STING to bind PI4P.

Interestingly, these residues have been recently linked to STING-associated vasculopathy with onset in infancy (SAVI)-causing STING mutations[52,53], these mutations promote the upregulation of anti-viral type-I-interferon stimulated genes (ISGs). C206Y, R281Q, and R284S all residing within the polymerization interface of STING and have been reported to be hyperactivation mutants independent of cGAMP or cGAS binding[54]. This region is separate from the aa153–177 sequence

that participates in STING dimerization[55,56]. It is unlikely then that aa280–295 is involved in dimerization, but instead critical for ER retention or interactions with a partner that regulated STING dimerization and activation[53,54]. This could be STIM, which has been shown to be a negative regulator of STING by sequestering it to the ER[30].

Based on our findings, we propose that upon HRV infection, HRV-2B protein affects the intracellular ER Ca$^{2+}$ homeostasis thus the STING–STIM1 interaction is disrupted (Supplemental Fig. 7). STING is released and assumes an "active state". Although STING assumes an active conformation, it does not polymerize to a level that leads to signal transduction, since a certain level of STING has to be occupied by cGAMP in order to achieve canonical activation[57]. The existence of a threshold of activation (all or none behavior[58]) is a central mechanism for STING to distinguish between self and foreign dsDNA, therefore its release from STIM allows it to assume an active state without dimerizing and reaching the threshold for full activation. In this active state, R281 and R284 are exposed and able to bind PI4P, which in turn promotes the migration of STING from the ER to the viral ROs, where it serves as a scaffold.

Since this translocation inhibits STING from performing its canonical function following HRV infection, it is possible that HRV infection may interfere with response against DNA viruses as the c-GAS-STING pathway will be impaired, although other innate immune sensors might compensate DNA sensing in these cases.

Following viral replicative infection, we observe the biogenesis of large autophagosomal like vesicles distinct from ROs with different lipid composition, enriched with phosphatidylserine (PS) lipids and autophagosomal markers which provide a cellular exit of the virus and are important for HRV assembly and transmission. STING traffics to these autophagosomal vesicles after viral replication and it is possible that STING, since it binds PI4P, could mediate the PI4P/PS lipid exchange between organelle membranes thus helping to maintain membrane lipid homeostasis. It is likely that the autophagosomal like organelle originated from ER-derived RO membranes.

This mechanism would fit with the primordial function of the STING-cGAS pathway to induce autophagy. It has been shown that a unique and important feature of the STING pathway is to activate autophagy independently of TBK-1 and interferon induction[22]. In this mechanism, STING translocates from the ER to the ERGIC which serves as a membrane source for LC3 lipidation and autophagosome biogenesis that target DNA viruses for the clearance from the cytosol to the lysosome. Thus, autophagy induction is an ancient, highly conserved function of the cGAS-STING pathway that is independent of the induction of interferon and inflammatory cytokines—similar to the mechanism that we have observed with HRV.

HRV has evolved to exploit STING and subvert the autophagic pathway and use it as an integral component of the viral life cycle without triggering the interferon or inflammatory response via STING. Our data indicate that HRV can exploit lipid homeostasis contributing to the generation of intracellular membranes and organelles potentially maximising virus replication and spread.

In conclusion our results identify STING as an important facilitator of HRV replication contributing to the assembly and exchange of different lipid microenvironments that leads to the generation of different type of organelles essential for HRV propagation (Fig. 7j).

This non-canonical role of STING has therapeutic potential and is particularly important not only for HRV replication but possibly for other RNA viruses as well, such as Coronaviruses which utilise replication organelles in order to facilitate their replication. In the future, targeting STING may be useful to improve preventive and effective treatment strategies to limit the overall burden of RV disease, and the consequent risk of developing chronic respiratory morbidity and mortality.

## Methods

***Tmem173* (Sting) tissue expression data**. The GTex data presented in this manuscript were obtained directly from the GTEx Portal, release v8 (dbGaP accession number phs000424.v8.p2). The Genotype-Tissue Expression (GTEx) Project was supported by the Common Fund of the Office of the Director of the National Institutes of Health, and by NCI, NHGRI, NHLBI, NIDA, NIMH, and NINDS.

**Acyl-biotin exchange (ABE) assay for palmitoylation**. Detection of Protein Palmitoylation by Immunoprecipitation and Acyl-Biotin Exchange (ABE) was performed according to Brigidi et al.[59]. Briefly, STING was immunoprecipitated from protein samples using 1 μg of anti-STING rabbit polyclonal antibody (Thermo Fisher, PA5-23381, 1:100) the complex was pulled using Protein A Sepharose AB beads (Thermo Fisher). To block non-palmitoylated cysteine residues, immune-precipitated STING was treated with 50 mM N-ethylmaleimide (NEM) in RIPA buffer (Sigma Aldrich, R0278) at 4 °C for 2 h on a shaker followed by three washes with RIPA buffer. STING was then treated with hydroxylamine (HAM) buffer to remove palmitate from Cysteine residues (1 M Hydroxylamine, 50 mM tris, 150 mM NaCl, 5 mM EDTA, 0.2% TX100, and pH 7.4) at room temperature for 2 h on a shaker. Then STING was treated with 4 μM of BMCC-Biotin in 50 mM Tris, 150 mM NaCl, 5 mM EDTA, 0.2% TX100, pH 6.2 for 1–2 h followed by three washes to remove excess biotin. Sixty microliter of reducing protein sample buffer was added and samples were boiled for 2 min to elute STING from beads. Twenty percentage of sample was loaded on to SDS PAGE gel. Membrane was blocked with 5% Bovine serum albumin (BSA) for overnight followed by incubation with streptavidin conjugated with HRP (Thermo Fisher, N100) at 1:30,000 for 60 min and Biotin-streptavidin HRP complexes were visualized by exposing the membrane to ECL and then to X-ray film.

**Antibodies**. Anti-STING rabbit polyclonal antibody (Invitrogen, PA5-23381, 1:100 IF)), anti-STING goat antibody (SantaCruz, sc241046, 1:100 iF)), J2 mAb English & Scientific Consulting, 1:100 IF. PI4P IgM mAb Echelon, ZP004-2, 1:100 IF), Anti Calreticulin goat polyclonal (ThermoFisher, PA1-33045, 1:100 IF). ERGIC-53 monoclonal antibody (ENZO, OTI1A8, 1:100 IF) as well as ERGIC-53 monoclonal antibody (Sigma-Aldrich, E1031, 1:100 IF). Alexa Fluor-488 or Fluor-546 labeled secondary antibodies against mouse or rabbit IgG or IgM (Thermofisher Scientific, 1:100 IF) were used.

**Agonists, antagonists, and lipids**. 2′3′-cGAMP was purchased from Invivogen. 2′3′-cGAMP (5 μg/ml) (Invivogen, tlrl-nacga23-02) was added to the cells complexed with LyoVec (Invivogen, Lyec-12) in order to aid internalization. ISD (Interferon stimulatory DNA)/Lyovec (Invitrogen, tlrl-isdc, 1 μg) was also used. Membrane lipid strips were obtained from Echelon. The following inhibitors were used: Fura-2 AM, Ca$^{2+}$ selective fluorescent indicator (Abcam, ab120873), BX795 (TBK-1 inhibitor, Invivogen, tlrl-bx7, 1 μM), RU.521 (cGAS inhibitor, Invivogen, inh-ru521, 500 nM). GSK2998533 (100 nM) & GSK2683449 (100 nM) were provided by GlaxoSmithKline (GSK). PIK93 was also used (Sigma Aldrich, 1 μM)

**Cell culture/viruses**. Bronchial epithelial cells (BEAS-2B, ATCC cat. Number CRL-9609) were cultured in RPMI medium containing 10% FBS, and 1% non-essential amino acids. Bronchial epithelial cells were seeded in lab-tek 8-well slides (80,000 cells/well) for two days. Once they were ~80% confluent, they were incubated with HRV (MOI:5) in 500 μl of serum-free medium (SFM) for 2 h at 37 °C. Following the stimulation, the supernatant was removed and the cells were washed ×2 with PBS, followed by fixation using 500 μl per well formalin for 15 min at RT.

Air liquid interface (ALIs) of airway epithelial cells were grown on transwells from normal donors (Epithelix, NHBE—Human Bronchial Epithelial Cells, Donor 28060 p3, and Donor 25265COPD Donor 63 p3) and cultured in Pneumacult media (Seeding density: 150,000 cells/well; Age of cells used in experiments post air-lift: 14 days. No differences were observed when cells were infected either through the apical or basolateral membrane suggesting the polarization of the cells had no effect.

Human rhinovirus A2 (HRV-A2) (ATCC, Cat. Number VR-482), HRV-B4 (ATCC, Cat. Number VR-284), HRV-B14 (ATCC, Cat. Number VR-284), HRV-A1B (ATCC, Cat. Number VR-1645), HRV-A16 (ATCC, Cat. Number VR-283) were purchased from the ATCC. In addition, HRV-A1B and HRV-A16 were also purchased from Virapur. Full-length HRV-C (HRV-C15) (GenBank accessionno.GU219984) was synthesized based on the published sequence[60] as described in ref. [61].

HRV RNA was quantified using primers in Supplemental Table 1.

These primers are complementary to the 5′ non translated region of HRV. RT-PCRs were performed in triplicate for each sample and quantified on an ABI 7000 real-time

PCR system (Applied Biosystems, CA). Data were analysed using GraphPad Prism (GraphPad Software, Inc.).

To generate viral stocks, 95% confluent HeLa-Ohio cells were infected and incubated at 34 °C/5%CO$_2$ for 3 days. All infected cells underwent freeze-and-thaw cycle three times, cell debris was removed by centrifugation at $500 \times g$ for 5 min at room temperature, supernatant was collected and stored at −80 °C. Virus titre was determined by plaque assay.

**Computational modelling of STING-PI4P interactions**. Potential PIP4 binding site identified between STING N-terminus and alpha3 helix with MOE Site Finder software (Molecular Operating Environment (MOE), 2019.01; Chemical Computing Group ULC, 1010 Sherbooke St. West, Suite #910, Montreal, QC, Canada, H3A 2R7, 2020) utilizing CryoEM STING structures 6NT6 and 6NT7[62].

A homology model of human STING was constructed using MOE.2019.0101 starting with on chicken STING (6NT7). The C-terminal domain (CTD) from structure 6DXG (internal PSILO code 1ASRC) was superimposed on the homology model. Residues Ile165 to Val343 were removed from the original model and replaced with those of 6DXG. To make a cleaner figure to show the alpha3 helix, the human full length cryoEM STING (6NT5) CTD was replaced with the modelled 6DXG one.

A conformation for PI4P was built starting with a ligand from 3MTC. Missing atoms were added, and a conformational search was carried out in MacroModel in Maestro 2020.1 (MCMM, 100,000 steps, 8 kcal/mol cut-off, other settings default). A low energy conformation of PI4P was then placed in the SiteFinder identified pocket near R281/284 of the full-length human STING homology model. Molecular dynamics was carried out for 100 ns in a 10A box of water using default conditions with Desmond in Maestro 2020.1.

**Confocal microscopy**. After fixation, the cells were washed ×2 with PBS and were permeabilised using 500 μl per well of PBS/0.02% BSA/0.02% Saponin and labeled with antibodies specific for PI4P, STING, J2 mAb to detect viral dsRNA, ERGIC (1:100) as well as secondary antibodies conjugated to the appropriate fluorophore (1:100). Following the secondary antibody incubation, the cells were washed three times using PBS/0.02% BSA/0.02% NaN$_3$. Finally all liquid was removed, as well as the plastic inserts. The cells were mounted with Vectashield, covered with a coverslip, and sealed using clear nail varnish.

Cells were imaged on a Carl Zeiss, Inc. LSM710 ELYRA P1 confocal using a 1.4 NA 63× Zeiss objective. The images were analysed using LSM 2.5 image analysis software (Carl Zeiss, Inc.).

No fluorescence was observed from an Alexa 488-labeled specimen using the 594 filters, nor was 594 fluorescence detected using the 488 filter sets.

**Quantification of co-localization**. In order to quantify the degree of co-localization, we used Costes' approach (Bolte and Cordelieres, 2006). Coste's approach, Pearson's correlation coefficients and p-values were calculated using MBF ImageJ with JACoP (Just Another Colocalisation Plugin) (http://macbiophotonics.ca/). This allows for the calculation of Pearson's correlation coefficient $R$(obs), which also accounts for any random overlay of pixels by generating the mean correlation coefficient $R$(rand) between $n$ images that have identical average pixel intensity to the original images, but a random distribution of pixels. $R$(obs) is not sensitive to background and has a linear regression range of −1 to 1, with −1 being total negative correlation, 0 being a random correlation and 1 being total positive. In theory, an $R$(obs) of +1 would represent a perfect pixel correlation between two channels in an image. Values greater than 0.5 are considered significant co-localization. In addition, Costes' randomisation method calculates the statistical significance of the Pearson's correlation coefficient. It returns a significance (p-value) expressed as a percentage.

**CRISPR STING knockout**. The following gRNA sequences were designed (GENscript) to uniquely target genes within the human genome. The gRNA sequences that were used with CRISPR/Cas9 knockout (KO) plasmid, to introduce a DSB for genome editing are shown in Supplemental Table 2.

Bronchial epithelial cells were seeded at a density of $0.8–3.0 \times 10^5$ cells/ml in 12-well plates. Fifty nanomolar guide RNA and 2.5 μg plasmid DNA encoding Cas9 were transfected to cells via *Trans*IT-X2 (MIRUS LLC, Madison, USA) following manufacturer's recommendations.

**Generation of STING mutants**. Human STING gene subcloned in pUNO1 (Invivogen) was used to generate the STING mutants. Point mutants were generated using standard PCR generated using the Geneart Quickchange Site-Directed Mutagenesis Kit (Invitrogen) according to the manufacturer's instructions.

**Phosphatidylserine vesicle isolation**. Cells were infected with HRV for 30 min and replaced with fresh serum free media for another 6–8 h. Extracellular media then was collected from virus-infected cells. Media was spun down at $500 \times g$ for 5 min (4 °C). Supernatant was collected and spun down again at $5000 \times g$ for 10 min (4 °C). Supernatant and pellet (vesicles) were collected for infection. Collected vesicles and free virus (either collected from supernatant post $5000 \times g$ spin

or from directly freeze-thawing [3×] the vesicles) can be used at this stage to infect cells. Further enrichment of PS vesicles from the pellet fraction was done with magnetic isolation using the Annexin V microbead kit as described (Milteny Biotec, CA). The magnetic microbeads lacking Annexin V were used to control for nonspecific binding.

**Fura-2/AM measurements**. BEAS-2B cells were transfected with 2 μg of p2B-myc using Lipofectin (Invitrogen). After 24 h, $5 \times 10^4$ cells were seeded onto 24 mm coverslips and grown for another 16 h before calcium measurements. Alternatively, BEAS-2B cells were infected with HRV1B (MOI 5) for 12 h.

The fluorescent Ca$^{2+}$ indicator Fura-2 was used to measure [Ca$^{2+}$cyt] at the single cell level as described[33,63]. Briefly, cells either infected with HRV or transfected with p2B-myc were incubated in medium supplemented with 2.5 μM Fura-2/AM for 30 min, washed with HT buffer to remove the extracellular probe, supplied with preheated HT buffer (supplemented with 1 mM CaCl$_2$), and placed in a thermostated (37 °C) incubation chamber on the stage of an inverted fluorescence microscope. Fluorescence was measured every 2 s with the excitation wavelength alternating between 340 and 380 nm and the emission fluorescence being recorded at 492 nm. At the end of the experiment, a region free of cells was selected, and one averaged background frame was collected at each excitation wavelength for background correction. To measure the [Ca$^{2+}$] of the stores, the amount of thapsigargin-releasable Ca$^{2+}$ was determined. After recording the cells in Ca$^{2+}$-containing HT buffer, the medium was replaced with Ca$^{2+}$-free HT buffer supplemented with 0.5 mM EGTA. After 10 min, the cells were challenged with 1 μM thapsigargin, and [Ca$^{2+}$]cyt was measured as described above.

**Statistical analysis**. Data represent the mean ± SD. Differences between groups were analysed using an unpaired student's $t$-test or 1-way ANOVA test as required, using GraphPad Prism 7 software, unless indicated otherwise. *$p < 0.05$, **$p < 0.01$, ***$p < 0.001$, ****$p < 0.0001$. For all experiments $n$ number of independent experiments, yielding consistent results.

**Reporting summary**. Further information on research design is available in the Nature Research Reporting Summary linked to this article.

## Data availability

All relevant data are available from the authors. Source data are provided with this manuscript. Source data are provided with this paper.

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

## Acknowledgements

This work was supported by the GlaxoSmithKline Immunology Network.

## Author contributions

K.T. and M.T. were involved in the conception and design of the work as well as the interpretation; K.T., M.T., L.B., G.J.D., K.S., N.N., and H.K. were involved in the acquisition, and analysis of the data; E.K., J.R., G.S.P., L.K.M., and J.B. provided reagents; M.T. supervised the project and wrote the original manuscript with input from all authors.

## Competing interests

The authors declare no competing interests.
