## [Peer Review File · Nature Communications]

Reviewers' Comments:

Reviewer #1:

Remarks to the Author:

The authors addressed the role of STING in replication of rhinovirus species. This is an elegant study using several approaches, the conclusions are well supported by the data and the findings have marked consequences for both potential intervention and, I guess, also for the response to DNA viruses. This and a couple of other issues would need attention.

Major comments:

1. There have been several other papers that have implicated STING in antiviral responses to RNA viruses. Short summary of these can be found in PMID: 28073693. This should be mentioned in the introduction.
2. The authors apparently use 'infection' and 'replication' as interchangeable terms (e.g. paragraph 107-117) which is not correct, or use 'replicative infection'. Throughout manuscript. E.g. line 183 should read replicative cycle instead of infectious cycle.
3. That STING cannot perform its canonical function following HRV infection (Figure 2F) is an interesting finding and although not mainstream, still deserves to be discussed. Do the authors assume that prior HRV infection may interfere with responses to DNA viruses, or does it lead to reactivation of latent DNA viruses. Is there any clinical data that can provide a clue?
4. It was not clear to me what was the source of bronchial/airway epithelial cells and whether there were differences in polarization between the BEAS 2B cells and bronchial epithelial cells. Did the polarization status affect the behavior of STING? If so, this needs to be discussed.

Minor comments:

1. The text between line 138 and 157 puts you somewhat in the wrong direction. It seems that the authors want to address whether HRV induces IFN beta production, but they really sort out whether STING and the cGAS pathway are involved. So, I would propose to slightly edit this paragraph.
2. Please clarify SAVI-causing STING mutations (line 434).
3. I guess Rq/4HK (vertical axis Figure 1B/D) is HRV replication. The latter is more self-explanatory.
4. Line 104,108, 203, 219, 259, 263, 315, 421, 441, 479, 485 has typos

Reviewer #2:

Remarks to the Author:

Comment to Authors:

The Stimulator of interferon genes (STING) is an ER-associated adaptor protein downstream of cGMP-AMP synthase (cGAS) in the innate immune cytosolic DNA-sensing pathway and is typically associated with antiviral effects through STING-mediated type 1 interferon production. In this manuscript, Triantafyllou M. and colleagues studied the non-canonical roles of STING in human rhinovirus (HRV) replication. The authors identified STING as a pro-viral host protein for HRV replication in a genome-wide siRNA screen in airway epithelial cells and validated the requirement of STING for the replication of all three HRV species HRV-A, HRV-B, and HRV-C in STING-deficient cells. The subcellular translocation of STING was carefully examined in HRV-infected cells. During the early stage of the HRV lifecycle, STING was released from the STIM1 in ER due to HRV-2B-mediated ER Ca²⁺-reduction, subsequently bound to phosphatidylinositol 4-phosphate (PI4P), a component of replication organelles, and then translocated to replication organelles. Further, STING was translocated to autophagosome-like vehicles at the late stage of the HRV life cycle which may facilitate virus assembly and release.

This is a surprising finding which could help us understand the roles of STING in RNA virus replication, and provide great insights on the development of novel approaches for anti-viral treatment. My concerns and comments are listed as following:

Major comments:

1. The finding that STING deficiency impaired HRV replication is convincing, while it is not clear

which step (s) of the HRV lifecycle requires the STING: genome replication, viral protein translation, assembly, and/or virion release? In Figure 6H, the authors showed decreased viral RNA in HRV-infected ALI cultures in the presence STING antagonist. However, the replication was examined 24 hours post virus infection, and the data is not sufficient to analyze which step of the HRV lifecycle was inhibited.

Besides, the current study showed STING is required for the replication of all three RV species A, B, and C. However, a recent study by K.L. McKnight (Proc. Natl. Acad. Sci. U. S. A. 2020. 117:27598-27607) showed that STING is required for RV-A and RV-C but not for RV-B. The authors need to comment on the contradictory results. In addition, is STING required for other enteroviruses, such as EV-D68?

2. As shown in Figure 3B, STING colocalizes with dsRNA and PIP4 with a punctuated pattern in HRV-A1B-infected BEAS-2B cells. Why the colocalization pattern in Figure 3D looks different from Figure 3B? The authors should provide a clear image to show colorization in replication organelles in RV-C15 infected cells.

3. Page 9, lines 205-216. The rationale for examining phosphatidylinositol 4-kinase III β (PI4KIII β) effects isn't clear. The roles of PI4KIII β in replication organelles during the infections of enteroviruses and rhinoviruses have been extensively studied (Dorobantu C.M., et al. 2015. J Virol; Lyoo H., et al. 2019. mBio; Roulin P.S., et al. 2018. J Virol.). It is not surprising to see reduced HRV replication in the presence of PI4KIII β inhibitors. The authors may want to remove Figure 3E and 3G to make the context concise.

It is more of interest to determine if PI4KIII β is required to recruit STING to the replication organelles and if PI4KIII β interacts with STING.

4. It has been extensively studied that PI4KIII β is recruited to the viral replication site by viral 3A protein during the infection of enteroviruses and rhinoviruses (Dorobantu C.M., et al. 2015. J Virol; Lyoo H., et al. 2019. mBio; Roulin P.S., et al. 2018. J Virol.), and subsequently exert its function to generate PI4P enriched replication organelles. Although the authors showed the interaction of STING with PI4P in Figure 5, the question of how STING was exported from the ER and recruited to the replication organelles wasn't well answered.

The authors suggested the migration of STING from ER to replication organelles required firstly the disassociation between STIM1 and STING and then the interaction between STING with PI4P. However, as shown the Figures S4, PI4P present in the replication organelles starting at the early stage of HRV infection. Where do the STING and PI4P interaction initially happen? In ER? The authors only showed two STING mutants (4posA and 4posE) that don't bind to PI4P, maintained in ER following HRV infection (Figure S3), I don't see any direct evidence showing a PI4P-STING interaction in ER. The authors should isolate the ER of HRV-infected cells and determine if there is STING-PI4P interaction.

The authors also should add lipid array results for the two STING mutants 4posA and 4 posE in Figure 5A. The authors also should determine show 1) if there is the formation of replication organelles; and 2) if PI4P translocate to replication organelles in these STING mutants expressing cells? Also please add the wildtype STING result in Figure S3A as control.

5. The authors showed the presence of STING in autophagosomal like vesicles at the late stage of HRV-infection, and the autophagy inhibitors bafA1 and 3-MA attenuated HRV infection. However, there is no sufficient data to prove that STING is required for the formation of autophagosomal like vesicles.

In Figure 6H, the authors gave the ALI culture with STING antagonist, H151. What is the rationale to determine the viral copies 24 hours post-infection?

The authors should give the H151 at the middle of the HRV lifecycle and quantify the extracellular HRV 8-hour post-infection to answer if STING plays a role in virus package and release.

6. The Result section includes many results and conclusions from previous studies. The authors may want to shorten the manuscript to aid communication of the most important findings.

Minor comments:

1. Please specify the "Rq/4HK" in the Method and figure legend.

2. Page 23, lines 555-558: are the primers HRV-C15 Fw and Rv used to determine HRV-C genome

only or for all RV strains?

3. Page 5, line 114. Please add the species to each indicated HRV strains. For example, change HRV1B to HRV-A1B.

4. What the difference between the upper and lower panels of FigureS3A?

5. Figure S3C should be Figure S3B.

Response to reviewers' comments:

The authors addressed the role of STING in replication of rhinovirus species. This is an elegant study using several approaches, the conclusions are well supported by the data and the findings have marked consequences for both potential intervention and, I guess, also for the response to DNA viruses. This and a couple of other issues would need attention.

Major comments:

1. There have been several other papers that have implicated STING in antiviral responses RNA viruses. Short summary of these can be found in PMID: 28073693. This should be mentioned in the introduction.

We agree with the reviewer and have added information on the involvement of STING in antiviral responses to RNA viruses in the introduction of the revised manuscript.

2. The authors apparently use 'infection' and 'replication' as interchangeable terms (e.g. paragraph 107-117) which is not correct, or use 'replicative infection'. Throughout manuscript. E.g. line 183 should read replicative cycle instead of infectious cycle.

We agree with the reviewer and have amended the use of infection/replication to replicative infection.

3. That STING cannot perform its canonical function following HRV infection (Figure 2F) is an interesting finding and although not mainstream, still deserves to be discussed. Do the authors assume that prior HRV infection may interfere with responses to DNA viruses, or does it lead to reactivation of latent DNA viruses. Is there any clinical data that can provide a clue?

We agree with the reviewer that the fact that STING can not perform its canonical function is an interesting finding and it deserves to be discussed. We think that prior HRV infection may interfere with responses to DNA viruses, although other DNA sensors might be compensating in this case; and have added this in the discussion of the revised manuscript. Unfortunately, there are no clinical data that might provide a clue.

4. It was not clear to me what was the source of bronchial/airway epithelial cells and whether there were differences in polarization between the BEAS 2B cells and bronchial epithelial cells. Did the polarization status affect the behavior of STING? If so, this needs to be discussed.

The bronchial/airway epithelial cells were from different donors obtained commercially from Epithelix. The polarization status of the cells did not affect the behaviour of STING and we have mentioned this in the materials and methods section of the revised manuscript.

Minor comments:

1. The text between line 138 and 157 puts you somewhat in the wrong direction. It seems that the authors want to address whether HRV induces IFN beta production, but they really

sort out whether STING and the cGAS pathway are involved. So, I would propose to slightly edit this paragraph.

We agree with the reviewer and have attempted to slightly edit the paragraph in the revised manuscript.

2. Please clarify SAVI-causing STING mutations (line 434).

STING-associated vasculopathy with onset in infancy (SAVI)-causing STING mutations, are mutations that promote the upregulation of anti-viral type-I-interferon stimulated genes and lead to STING hyper-activation. We have added this information in the discussion when we mention SAVI-causing STING mutations in the revised manuscript.

3. I guess Rq/4HK (vertical axis Figure 1B/D) is HRV replication. The latter is more self-explanatory.

We agree with the reviewer that it is not clear what Rq/4HK is, it is HRV replication and we have amended the axis to the figures in order to clarify this.

4. Line 104,108, 203, 219, 259, 263, 315, 421, 441, 479, 485 has typos

We agree with the reviewer and have amended all the typos.

Reviewer #2 (Remarks to the Author):

Comment to Authors:

The Stimulator of interferon genes (STING) is an ER-associated adaptor protein downstream of cGMP-AMP synthase (cGAS) in the innate immune cytosolic DNA-sensing pathway and is typically associated with antiviral effects through STING-mediated type 1 interferon production. In this manuscript, Triantafyllou M. and colleagues studied the non-canonical roles of STING in human rhinovirus (HRV) replication. The authors identified STING as a pro-viral host protein for HRV replication in a genome-wide siRNA screen in airway epithelial cells and validated the requirement of STING for the replication of all three HRV species HRV-A, HRV-B, and HRV-C in STING-deficient cells. The subcellular translocation of STING was carefully examined in HRV-infected cells. During the early stage of the HRV lifecycle, STING was released from the STIM1 in ER due to HRV-2B-mediated ER Ca²⁺reduction, subsequently bound to phosphatidylinositol 4-phosphate (PI4P), a component of replication organelles, and then translocated to replication organelles. Further, STING was translocated to autophagosome-like vehicles at the late stage of the HRV life cycle which may facilitate virus assembly and release.

This is a surprising finding which could help us understand the roles of STING in RNA virus replication, and provide great insights on the development of novel approaches for anti-viral treatment. My concerns and comments are listed as following:

Major comments:

1. The finding that STING deficiency impaired HRV replication is convincing, while it is not clear which step (s) of the HRV lifecycle requires the STING: genome replication, viral protein translation, assembly, and/or virion release? In Figure 6H, the authors showed decreased viral RNA in HRV-infected ALI cultures in the presence STING antagonist. However, the replication was examined 24 hours post virus infection, and the data is not sufficient to analyze which step of the HRV lifecycle was inhibited.

We agree with the reviewer and in order to determine in which steps of the HRV replicative cycle STING is involved, we added H-151 (STING antagonist) before as well as in the middle of the HRV replicative cycle (3 hrs) and then assessed the extracellular HRV at 8 h post infection (as the reviewer has suggested). We found that H-151 could inhibit HRV replication if administered prior to infection, and when administered in the middle of the replicative cycle (3hr) it diminished the extracellular HRV 8 hr p.i., suggesting that STING also plays a role in the packaging and release of the virus. Therefore, based from our findings, we propose that STING contributes to the formation and function of two different organelles during the HRV replicative cycle, 1) the replication organelles (ROs), required for replication, and 2) the autophagosomes that play a role in the package and release of HRV.

Besides, the current study showed STING is required for the replication of all three RV species A, B, and C. However, a recent study by K.L. McKnight (*Proc. Natl. Acad. Sci. U. S. A.* 2020. 117:27598-27607) showed that STING is required for RV-A and RV-C but not for RV-B. The authors need to comment on the contradictory results. In addition, is STING required for other enteroviruses, such as EV-D68?

We agree with the reviewer that a recent study has been published demonstrating that STING is required for HRV-A and HRV-C but not HRV-B replication. This is partly in agreement with our work. We have showed that STING is required for all HRV strains, including HRV-B. The discrepancy between the two studies is probably due to the fact that the McKnight et al study mainly uses cell lines, which are not biologically relevant for HRV infection and thus the tissue tropism of the different HRV strains might not have been relevant in those cell types. They mainly used Huh (hepatic) cells as well as HeLa (cervical epithelial) cells for their study, whereas we have used bronchial epithelial cells as well as air-liquid interface (ALI) primary bronchial cells, which are the main tissues that all HRV strains infect. Furthermore, the McKnight et al study only utilises only one HRV-B strain, HRV-B14, in order to determine whether the HRV-B group viruses are STING-dependent. In our study, we have utilised HRV-B14 as well as HRV-B4 (and several other B strains – data not shown) in order to confirm the finding that HRV-B group are STING-dependent. It is possible that the strain that they are using have some deficiency and the finding should be tested using other HRV-B strains as well – just as we have done in our study.

In addition, the McKnight et al study although demonstrates that STING is required for HRV replication, it does not reveal the molecular mechanisms involved, therefore our study is advancement to the field describing the molecular mechanisms involved.

Regarding, EV-D68, since it was previously classified as a Rhinovirus, we would predict that it could utilise STING for its replication. Unfortunately, due to the coronavirus pandemic and the limited time that we had access to the labs, we were unable to perform experiments with EV-D68. In the McKnight et al study they have tested EV-D68 and claim that it does not require STING for its replication, but due to the tissue tropism mentioned above, we feel that this observation will need revisiting in a future publication.

2. As shown in Figure 3B, STING colocalizes with dsRNA and PIP4 with a punctuated pattern in HRV-A1B-infected BEAS-2B cells. Why the colocalization pattern in Figure 3D looks different from Figure 3B? The authors should provide a clear image to show colorization in replication organelles in RV-C15 infected cells.

We agree with the reviewer that the colocalization pattern in Figure 3D looks different from Figure 3B. The reason for this is because Figure 3D is from air-liquid interface (ALI) bronchial cells, whereas Figure 3B depicts BEAS-2B cells. Unfortunately, HRV-C15 only grows on ALIs *in vitro* and ALI cells are grown in inserts and can only be imaged as a dense monolayer (cut from the insert) thus we can not provide a clearer image in this case. The image does show the colocalization between STING and replication organelles, albeit without very high resolution. In contrast, Figure 3B depicts BEAS-2B cells where we can control the density of the cells and grow them on a lab-tek microscope slide thus it is easier to provide a clearer image of colocalization.

3. Page 9, lines 205-216. The rationale for examining phosphatidylinositol 4-kinase III β (PI4KIII β) effects isn't clear. The roles of PI4KIII β in replication organelles during the infections of enteroviruses and rhinoviruses have been extensively studied (Dorobantu C.M., et al. 2015. *J Virol*; Lyoo H., et al. 2019. *mBio*; Roulin P.S., et al. 2018. *J Virol*). It is not surprising to see reduced HRV replication in the presence of PI4KIII β inhibitors. The authors may want to remove Figure 3E and 3G to make the context concise. It is more of interest to determine if PI4KIII β is required to recruit STING to the replication organelles and if PI4KIII β interacts with STING.

We agree with the reviewer that the role of PI4KIII β and enteroviruses has been extensively studied, but for completeness we wanted to include this in the manuscript. We have moved Figure 3E and 3G to the supplemental figures, in order to make the main manuscript and figures more concise.

We agree with the reviewer that it would be more of interest to determine if PI4KIII β is required to recruit STING to the replication organelles and whether PI4KIII β interacted with STING. Therefore, we have performed fluorescence resonance energy transfer (FRET) experiments, that can detect molecular interactions down to 20 nm, between STING and PI4KIII β , PI4P as well as ER. In addition, we have studied these interactions between ER and STING, PI4KIII β as well as PI4P. Our FRET data demonstrates that STING associates with PI4KIII β and PI4P in the ER within 1h of HRV infection. Therefore PI4KIII β does interact with STING and possibly plays a role in its recruitment from the ER to the ROs. We have included this new data in the revised manuscript.

4. It has been extensively studied that PI4KIII β is recruited to the viral replication site by viral 3A protein during the infection of enteroviruses and rhinoviruses (Dorobantu C.M., et al. 2015. *J Virol*; Lyoo H., et al. 2019. *mBio*; Roulin P.S., et al. 2018. *J Virol*), and subsequently exert its function to generate PI4P enriched replication organelles. Although the authors showed the interaction of STING with PI4P in Figure 5, the question of how STING was exported from the ER and recruited to the replication organelles wasn't well answered.

The authors suggested the migration of STING from ER to replication organelles required firstly the disassociation between STIM1 and STING and then the interaction between STING with PI4P. However, as shown the Figures S4, PI4P present in the replication organelles starting at the early stage of HRV infection. Where do the STING and PI4P

interaction initially happen? In ER? The authors only showed two STING mutants (4posA and 4posE) that don't bind to PI4P, maintained in ER following HRV infection (Figure S3), I don't see any direct evidence showing a PI4P-STING interaction in ER. The authors should isolate the ER of HRV-infected cells and determine if there is STING-PI4P interaction.

We agree with the reviewer that we hadn't demonstrated direct interaction of STING with PI4P. We have performed FRET experiments in order to demonstrate the direct association of STING with PI4KIII B as well as PI4P in the ER in the revised manuscript (Figure 3e). We have also performed specific FRET experiments examining the interaction of ER with STING and PI4P (Figure 3f). These experiments have demonstrated that STING interacts with PI4P within the ER within 1h of infection. We have added the new data in the revised manuscript.

The authors also should add lipid array results for the two STING mutants 4posA and 4 posE in Figure 5A. The authors also should determine show 1) if there is the formation of replication organelles; and 2) if PI4P translocate to replication organelles in these STING mutants expressing cells? Also please add the wildtype STING result in Figure S3A as control.

We agree with the reviewer and have performed confocal experiments in order to determine whether the STING mutants 4posA and 4posE 1) form replication organelles and 2) whether they co-localise with PI4P (revised manuscript, Figure S3a two bottom row of panels). The confocal experiments demonstrate that the two mutants are not able to form replication organelles (we don't see the punctate organelles that we see with wildtype) and they do not co-localise with PI4P. We have also performed experiments with the two mutants using the lipid array (revised manuscript, Figure 5a) and it has been clearly shown that the mutants are not able to bind PI4P in the lipid array.

In addition, we have added the wildtype STING result Figure S3a as a control, as requested by the reviewer.

5. The authors showed the presence of STING in autophagosomal like vesicles at the late stage of HRV-infection, and the autophagy inhibitors bafA1 and 3-MA attenuated HRV infection. However, there is no sufficient data to prove that STING is required for the formation of autophagosomal like vesicles.

In Figure 6H, the authors gave the ALI culture with STING antagonist, H151. What is the rationale to determine the viral copies 24 hours post-infection?

The rationale for examining viral copies 24h post infection, was to determine whether the STING antagonist, H-151, could inhibit viral replication.

The authors should give the H151 at the middle of the HRV lifecycle and quantify the extracellular HRV 8-hour post-infection to answer if STING plays a role in virus package and release.

We agree with the reviewer and have performed the experiment, where H-151 is administered in the middle of the HRV replicative lifecycle and quantified the extracellular HRV 8h post infection. The data suggests that when H-151 was administered in the middle of the HRV replicative lifecycle, the extracellular HRV at 8h was diminished, thus suggesting that STING plays a role in the virus package and release.

6. The Result section includes many results and conclusions from previous studies. The authors may want to shorten the manuscript to aid communication of the most important findings.

Minor comments:

1. Please specify the “Rq/4HK” in the Method and figure legend.

We agree with the reviewer and have specified what Rq/4HK stands for in the figure legend as well as the method (it stands for HRV replication).

2. Page 23, lines 555-558: are the primers HRV-C15 Fw and Rv used to determine HRV-C genome only or for all RV strains?

The primers in lines 555-558 have been used to determine the HRV-C genome. We have now added the primers that we used in order to detect all the other strains.

3. Page 5, line 114. Please add the species to each indicated HRV strains. For example, change HRV1B to HRV-A1B.

We agree with the reviewer and have added the species for all indicated HRV strains.

4. What the difference between the upper and lower panels of FigureS3A?

We agree with the reviewer that the lower panels of Figure S3a were repetitive and redundant and thus have removed the bottom panels and replaced them with confocal microscopy panels demonstrating the lack of interaction of the STING mutants with PI4P.

5. Figure S3C should be Figure S3B.

We agree with the reviewer and have amended the numbering of the figure

Reviewers' Comments:

Reviewer #1:

Remarks to the Author:

The authors have addressed my comments. I have read the responses to comment by reviewer 2 and have no further comments.

Reviewer #2:

Remarks to the Author:

The authors have carefully revised the manuscript and addressed my major concerns.

However, there are several minor comments as listed below:

1. Fig 1a, please add the species to each indicated HRV strain.
2. Figure 3g is supposed to be moved to supplemental, but I don't see it. And there is no Fig S1g in the supplemental.
3. No description is added for Figure 3e and 3f in the figure legend, and Figure 3f is cited in the result.
4. Page 13, sentence "we observed using confocal microscopy that both the STING 4posA and 4posE mutants did not translocate from the ER and thus could no longer interact with PI4P (Figure S3a)," In the revised manuscript, the authors showed STING-PI4P interaction within ER 1 hour post-infection. However, the sentence above sounds the interactions between PI4P and STING mutants were impaired because STING 4posA and 4posE mutants didn't translocate from the ER. The authors may want to revise this sentence.